# Breast Cancer Stem Cell Membrane Biomarkers: Therapy Targeting and Clinical Implications

**DOI:** 10.3390/cells11060934

**Published:** 2022-03-09

**Authors:** Inês Conde, Ana Sofia Ribeiro, Joana Paredes

**Affiliations:** 1i3S, Institute of Investigation and Innovation in Health, 4200-135 Porto, Portugal; iconde@ipatimup.pt (I.C.); aribeiro@ipatimup.pt (A.S.R.); 2Ipatimup, Institute of Molecular Pathology and Immunology, University of Porto, 4200-135 Porto, Portugal; 3Faculty of Medicine, University of Porto, 4200-319 Porto, Portugal

**Keywords:** breast cancer stem cells, cell membrane biomarkers, targeted therapies, translational oncobiology

## Abstract

Breast cancer is the most common malignancy affecting women worldwide. Importantly, there have been significant improvements in prevention, early diagnosis, and treatment options, which resulted in a significant decrease in breast cancer mortality rates. Nevertheless, the high rates of incidence combined with therapy resistance result in cancer relapse and metastasis, which still contributes to unacceptably high mortality of breast cancer patients. In this context, a small subpopulation of highly tumourigenic cancer cells within the tumour bulk, commonly designated as breast cancer stem cells (BCSCs), have been suggested as key elements in therapy resistance, which are responsible for breast cancer relapses and distant metastasis. Thus, improvements in BCSC-targeting therapies are crucial to tackling the metastatic progression and might allow therapy resistance to be overcome. However, the design of effective and specific BCSC-targeting therapies has been challenging since there is a lack of specific biomarkers for BCSCs, and the most common clinical approaches are designed for commonly altered BCSCs signalling pathways. Therefore, the search for a new class of BCSC biomarkers, such as the expression of membrane proteins with cancer stem cell potential, is an area of clinical relevance, once membrane proteins are accessible on the cell surface and easily recognized by specific antibodies. Here, we discuss the significance of BCSC membrane biomarkers as potential prognostic and therapeutic targets, reviewing the CSC-targeting therapies under clinical trials for breast cancer.

## 1. Introduction

Upon cancer stem cells (CSCs) first identification in leukaemia, in 1994 [1], many studies intended to unravel their key traits and mechanisms in cancer biology. Importantly, over the years, increasing evidence has revealed a role for these highly tumourigenic cancer cells in cancer progression and metastasis [2].

### 1.1. Cancer Stem Cells’ Biology and Their Role in Metastasis

The CSC theory postulates that tumour heterogeneity has a hierarchical cellular organization, driven by a small subpopulation of cancer cells showing stem-like properties and high tumourigenic potential, which is responsible for tumour formation and progression. The so-called CSCs or tumour-initiating cells (TICs) are characterized by their great plasticity and strong self-renewal ability, as well as by their multilineage differentiation potential [3,4]. Although these key properties are shared by CSCs and normal stem cells, the regulatory mechanisms that maintain the stem/differentiation balance are abolished; therefore, uncontrolled CSC proliferation ultimately leads to tumourigenesis [5].

The cell of origin for CSCs is still a matter of debate. However, it is known that it can vary among individual tumours. There are studies demonstrating that CSCs arise from somatic alterations in normal tissue stem/progenitor cells [3]; however, others suggest that CSCs derive from a pool of intermediary “transient amplifying” cells that are mitotically active and accumulate mutations to dedifferentiate and sustain the CSC pool [6,7]. In addition, CSCs may also arise due to the accumulation of genetic or epigenetic alterations, occurring in somatic differentiated cells. There are still studies suggesting a putative role of inflammation in the induction of a CSC state. For instance, Get et al. demonstrated that microenvironmental stress signals promoted the activation of stress-induced regulatory elements and the silencing of homeostatic signals, triggering the switch of stem cell fate in skin cancer [6,8]. Moreover, despite the cell of origin, it is accepted that there are several CSC pools with distinct properties within tumours, reflecting the increased levels of plasticity that are characteristic of these cells [6].

Similar to normal stem cells, CSCs are also commonly regulated by pluripotent transcription factors, including Oct4, Sox2, Nanog, KLF4, and MYC [5,6], as well as by intracellular pathways, such as Wnt, Notch, and Hedgehog (Hh) signalling [9]. Moreover, CSCs are also regulated by extracellular and microenvironmental factors, including cancer-associated fibroblasts (CAFs), extracellular matrix (ECM), exosomes, vascular niches, tumour-associated macrophages (TAMs), and hypoxia [5].

CSCs’ ability to self-renew and differentiate into multiple cell types confer them the capacity to initiate/drive tumourigenesis or to form/repopulate heterogeneous cancer cell populations, which are crucial for cancer progression. Generally, CSCs undergo asymmetric division, meaning that one CSC gives rise to another CSC with unlimited proliferative potential through self-renewal, and to a phenotypically distinct cancer cell with a limited lifespan which integrates with the tumour bulk through differentiation mechanisms [10]. However, CSCs can also divide symmetrically and self-renew, with two CSCs resulting from cell division, keeping the stem cell progeny. Symmetric division usually occurs to excessively increase tumour growth and repopulation as a response to stress conditions, such as cell loss during treatment [11] (Figure 1).

In solid tumours, the fraction of CSCs was accepted to represent a very small percentage of all cancer cells [12]. However, the identification of CSCs is of clinical relevance since it has significant prognostic and therapeutic implications. In fact, Samanta et al. reported that breast cancer cells displaying a BCSC phenotype are not responsive to chemotherapy and that paclitaxel treatment led to an increase in this BCSC phenotype [13]. Additionally, several lines of evidence demonstrate that BCSCs from different cancer cell line models are resistant to radiotherapy [14]. Thus, over the years, research efforts have been made to find new strategies to identify CSCs. Although the gold standard method to define CSCs consists of serial in vivo transplantation in animal models, CSCs can be also identified through the expression of several markers, mainly cell-surface proteins such as CD44, prominin-1 and EpCAM, for instance, whose expression is generally deregulated in CSCs, compared with their non-CSC counterparts [15]. In addition, in vitro tumoursphere formation assay is also used to identify CSC populations, since it evaluates the self-renewal and anoikis resistance capacity of cells and, therefore, is widely used in preclinical studies [16].

### 1.2. Cancer Stem Cells’ Resistance to Therapy

CSCs are highly resistant to traditional cancer therapies, including chemotherapy and radiation. This resistance is due to functional or molecular properties that are particularly relevant in CSCs. The mechanisms of CSCs’ resistance to therapy can be intrinsic, explained by genetic alterations leading to the aberrant expression of proteins, or extrinsic, resultant from microenvironmental cues, which largely contribute to tumour relapse and metastasis. Cellular quiescence, increased expression of ATP-binding cassette (ABC) transporters, detoxifying agents and antiapoptotic proteins, evasion to the immune system, deregulation of pathways associated with self-renewal, differentiation, apoptosis and survival, epigenetic reprogramming, enrichment in DNA damage response, and reactive oxygen species (ROS) scavenging and altered metabolism are examples of mechanisms contributing for CSCs’ resistance to conventional anticancer therapies [3,17].

The prime mechanisms by which conventional chemotherapies act are through the induction of DNA damage and apoptosis that target and eliminate highly proliferative cells [18]. CSCs are slow cycling and show the ability to be dormant for long periods of time until there are favourable conditions to their growth. Dormancy is an important feature for CSCs’ therapy resistance, since it is considered to be a quiescent state, where CSCs remain arrested in the G0 phase of the cell cycle, which allows them to be resistant to conventional chemotherapy [15]. Moreover, CSCs show increased expression of antiapoptotic proteins, as well as the ability to enhance their DNA damage response mechanisms. For instance, CSCs present high levels of phosphorylated DNA damage response proteins, increased activation of G2/M checkpoint, and more efficient double-strand break mechanisms in order to repair DNA damage induced by chemotherapeutic agents and radiation, thus avoiding apoptosis [19].

CSCs show aberrant increased expression of ABC transporters, which are multidrug resistance proteins that promote cellular efflux of the chemotherapeutic agents using ATP hydrolysis, thus protecting cancer cells from drug damage and inducing drug resistance [15]. CSCs show increased aldehyde dehydrogenase (ALDH) activity, which is a main responsible for resistance to chemotherapeutic drugs and radiation. ALDH promotes the metabolization of chemotherapeutic agents and eliminates oxidative stress, reducing chemotherapy toxic effects. Additionally, ALDH eliminates free radicals induced for ionizing radiation, stimulating radiation resistance [20]. In addition, CSCs possess more efficient ROS-scavenging mechanisms, which allow the prevention of cell damage induced by ROS [21]. For instance, it has been demonstrated that CD44^+^CD24^−^ BCSCs have the ability to reduce intracellular ROS levels (induced by radiation) through the overexpression of a radical scavenger by an enhancement of reduced glutathione, as well as increased NADH and FADH_2_ [22,23,24,25].

Another major factor involved in CSCs’ resistance is the deregulation of intracellular signalling pathways associated with self-renewal, differentiation, apoptosis, and survival, in particular, Notch, Hh, and Wnt signalling. CSCs frequently present abnormal deregulated Notch signalling, which ultimately leads to uncontrolled cell proliferation and decreased differentiation and apoptosis, contributing to CSC maintenance [5]. Increased activation of Wnt signalling has been shown to be very important to the maintenance of CSCs since it plays a key role in inducing the transformation of dormant CSCs into active CSCs to promote cell cycle progression. It also regulates dedifferentiation and inhibition of apoptosis in CSCs and mediates its role in CSC-driven metastasis [5]. It is believed that the activation of Hh signalling is also critical for self-renewal, growth, and metastasis of CSCs. In addition, activation of the Hh signalling pathway is associated with Wnt signalling dysfunction, further contributing to the support of the self-renewal ability of CSCs [17].

Although the metabolic traits, as well as epigenetic reprogramming systems of CSCs, are yet not fully understood, it seems that they might represent important mechanisms contributing to CSCs’ therapy resistance [26,27]. In fact, Magnani et al. demonstrated a connection between methylation events and resistance to endocrine therapies in oestrogen receptor-positive breast cancer [27]. Additionally, Chaffer et al. observed that basal-like breast non-CSCs keep Zeb1 (an epithelial-to-mesenchymal transition transcription factor involved in the generation of BCSCs) promoter in a poised configuration to easily respond to microenvironmental stimuli, maintaining cellular plasticity and promoting the transition of non-BCSCs into BCSCs [28]. Moreover, CSCs show several escape mechanisms, such as loss of expression of key molecules for tumour immunosurveillance (e.g., IFN gamma, IL12, TRAIL). Therefore, they are able to elude the immune system, hence considered immune-privileged cells, which significantly contribute to therapy resistance [29].

## 2. Breast Cancer Stem Cells (BCSCs)

In breast cancer, CSCs are referred to as breast cancer stem cells (BCSCs). In recent years, there have been significant improvements in prevention, early diagnosis, and treatment options, which resulted in a significant decrease in breast cancer mortality rates. However, despite these improvements, the high rates of incidence combined with therapy resistance result in cancer relapse and metastasis, contributing to the still unacceptably high mortality of breast cancer patients. In fact, there are studies describing BCSCs as key elements in therapy resistance, breast cancer relapse, and metastasis [2,5,15,25,30,31,32]. Accordingly, a high proportion of BCSCs within breast tumours have been demonstrated as being correlated with a poor outcome [33]. Therefore, BCSCs constitute an area of major concern in oncology, highlighting the need for the development of new therapeutic strategies that allow the overcome of therapy resistance for the benefit of breast cancer patients.

BCSCs were firstly described by All-Hajj et al. [34] in 2003. The authors observed that as few as a hundred CD44^+^CD24^−^ cells, which were further proved to be BCSCs, were able to form tumours when injected in the mammary fat pad of non-obese diabetic severe combined immunodeficient (NOD/SCID) mice, whereas a greater number of breast cancer cells with other phenotypes failed to sustain tumour growth. Moreover, they demonstrated that CD44^+^CD24^−^ single-cell suspensions showed self-renewal, clonal mammosphere formation, and chemotherapy resistance abilities, as well as extensive proliferation [34].

The great plasticity of BCSCs contributes to intra-tumoural heterogeneity, as well as to the existence of several BCSC pools within tumours, which represents a major challenge for tumour eradication. In fact, Liu et al. observed that phenotypic interconversion between epithelial and mesenchymal states can occur in BCSCs and that each state is associated with different localisation within the tumours, as well as with distinct proliferative and invasive capacities [35]. The transition between these two states is a dynamic and reversible process [36], whose regulation by epigenetic alterations or tumour microenvironment signalling occurring during tumour progression plays a critical role in promoting invasion and metastatic capabilities. Interestingly, epithelial-like BCSCs expressing the BCSC marker ALDH are more proliferative and are located at central areas of the tumour, whereas CD44^+^CD24^−^ mesenchymal-like BCSCs are more quiescent and located at the invasive front of the tumour. During the metastatic cascade, the mesenchymal CD44^+^CD24^−^ phenotype is crucial for the initial steps, as it is important for cells to be able to enter the circulation, as well as to seed and form micrometastasis; however, when the invasive edge becomes the interior of the tumour, the switch to the ALDH^+^ epithelial state is important for establishing macrometastasis. In addition to these two BCSCs populations, the authors also described the existence of a third CD44^+^CD24^−^ALDH^+^ population, which was associated with the greatest tumourigenic potential and invasion capability among the three BCSCs’ populations, and it was believed that this phenotype represents the most purified BCSC state [35].

Accumulating evidence suggests that BCSCs play key roles in metastasis. In fact, primary breast tumour cells with a CD44^+^CD24^−^ profile were associated with increased metastatic capacity and poor survival probability [37]. Moreover, the most direct evidence that CSCs can establish metastasis was demonstrated by Liu et al. using CD44^+^CD24^−^ BCSCs from breast tumour patient-derived cells show the ability to form tumours, as well as to spontaneously metastasize to the lungs in orthotopic mouse models [38]. In addition, Balic et al. demonstrated that most early disseminated cancer cells detected in the bone marrow of breast cancer patients have a putative breast cancer stem cell phenotype [39]. In further studies, using a xenograft model, Charafe-Jauffret et al. showed that ALDH^+^ cancer cells display increased metastatic abilities when compared with the ALDH^−^ cancer cells [40]. Notably, our group has recently demonstrated that breast cancer cells that have in vivo metastatic capacity in mice models [41] have increased CSC activity, but most importantly, we showed that human metastatic lesions [42], specifically brain metastasis, are enriched for a BCSC phenotype (determined by CD44, integrin subunit *α*6, EpCAM, cadherin-3, and ALDH1 expression). Importantly, we were able to validate a new in vivo tool based on the chick chorioallantoic membrane (CAM) model that can be used as an alternative, appealing, and novel approach in the CSC research field to study CSC activity. Importantly, this novel tool can also be applied to test the efficacy of anti-CSC drugs, with the potential to be used in a personalized medicine context [41].

### 2.1. BCSC Biomarkers

Although there are no universal markers that allow the specific recognition of BCSCs, this population is generally identified by the expression of the surface markers CD44, CD24, and ALDH1, which are the most accepted biomarkers associated with a BCSC phenotype. These markers have initially been defined by All-Hajj et al. [34] and further validated by several authors [43,44,45]. However, additional markers or different combinations of their expression are described to identify BCSC populations, including transcription factors commonly associated with normal stem cell functions (e.g., Sox2, Oct4, and Nanog) and cell surface markers such as integrin subunit *α*6, CD29, CD61, EpCAM, cadherin-3, HER2, prominin-1, and C-X-C motif chemokine receptor 1.

#### 2.1.1. Transcription Factors (TFs)

Although the majority of BCSC biomarkers are intracellular or membrane proteins, there are some pluripotency TFs, such as Sox2, Oct4, and Nanog, as well as epithelial-to-mesenchymal transition (EMT) TFs, such as Zeb1, that have been correlated with BCSCs’ origin. In particular, Oct4 is an embryonic stem cell marker, whose expression has been reported as being an independent prognostic factor in hormone-receptor-positive breast cancer associated with ALDH1 positivity, high proliferative ki67 index, and high histological tumour grade [46]. Additionally, Bhatt et al. demonstrated a possible correlation between Oct4 expression and breast cancer cells’ resistance to tamoxifen treatment [47]. Nanog is also an embryonic stem cell marker that has been described as upregulated in invasive breast cancer. Moreover, it has been directly correlated with tumour size, grade, and stage, as well as with lymph node metastasis and poor overall survival [48]. Sox2 is a pluripotency TF that has been reported as being highly expressed in breast cancer tissues [49,50]. Interestingly, Leis et al. observed that Sox2 is expressed and is necessary for in vitro mammosphere formation. Moreover, the authors observed that silencing Sox2 in breast cancer cells led to a delay in tumour formation in mice xenografts [51].

Increasing lines of evidence suggest a correlation between EMT and its associated TFs as ZEB1 and BCSCs formation. For instance, Jiang et al. demonstrated that ZEB1 knockout mouse-derived breast cancer cells present a decrease in the percentage of CD44^+^CD24^−^, as well as in the ALDH1^+^ population, accompanied by a reduction in the mammosphere formation capacity [52]. Moreover, increased ZEB1 expression in triple-negative breast cancer patients, the breast cancer molecular subtype with worse prognosis and with an enrichment in the CSC population, is predictive of radiotherapy relapse [53].

#### 2.1.2. Cell Surface Proteins

Membrane proteins are an area of special interest since they are accessible on the cell surface and are one of the first classes of proteins to be impacted under pathological conditions [54]. Importantly, membrane proteins constitute promising prognostic and therapeutic markers, because they are accessible targets for monoclonal antibodies and other drugs [55].

##### CD44 Molecule

CD44 is a transmembrane glycoprotein, whose important ligand is hyaluronic acid. It is involved in several biological processes that are important during tumour progression and metastasis—namely, cellular survival, proliferation, differentiation, and adhesion, and is a key mediator in the transendothelial migration of breast cancer cells during the metastatic cascade [56]. BCSCs are characterized by high CD44 expression levels, which are associated with the maintenance of a multipotent phenotype [57]. Regarding this, CD44 has been widely used to identify and isolate BCSCs, in addition to being considered a potential target for BCSCs-targeting therapies [24,58,59].

##### CD24 Molecule

CD24 or heat-stable antigen (HAS) is a sialoprotein involved in multiple biological processes such as cell adhesion, migration, proliferation, invasion, and metastasis [60]. Although CD24 is an accepted CSC biomarker, whether their expression (overexpression or lack of expression) is associated with stem-like properties and tumourigenic potential seems to be context-dependent, varying across tumour types [61]. However, in BCSCs, CD24 expression is usually very low or even absent. Further corroborating this CD24^−^ BCSC phenotype, evidence from in vitro studies demonstrated that the overexpression of CD24 was associated with the inhibition of stem-like properties in breast cancer cells [62]. In addition, another study suggested that CD24^−^ phenotype was associated with therapy resistance in breast cancer cell lines [63]. Generally, CD24 is used in combination with CD44, making the CD44^+^CD24^−^ phenotype a classical BCSC biomarker [60,64].

##### Integrin Subunit *α*6

Integrin subunit *α*6, also known as CD49f, is a type I transmembrane glycoprotein, belonging to the *α*-subfamily. Importantly, CD49f might assume two distinct cytoplasmic domains generated by mRNA alternative splicing variants, *α*6A and *α*6B, that show different stem-like functions. Specifically, it has been shown that the *α*6B variant is the isoform that closely associates with a CSC phenotype [65,66,67]. Usually, CD49f forms heterodimers with integrins *β*1 (CD29) and *β*4 (CD104), functioning as a receptor for several laminins and playing an important role in cell adhesion to the extracellular matrix (ECM) [68]. Furthermore, studies demonstrate that CD49f heterodimers cooperate with signalling pathways to induce stem-like and invasive properties to breast cancer cells. In breast cancer, high expression of CD49f is a poor prognostic marker associated with reduced survival [69].

##### Epithelial Cell Adhesion Molecule

Epithelial cell adhesion molecule (EpCAM) is a transmembrane glycoprotein expressed in the majority of epithelial cells and it mediates homophilic Ca^2+^ independent cell–cell adhesion [15], contrary to what is seen for epithelial cadherins. It is involved in several biological processes, including cell signalling [70], migration [71], proliferation, differentiation, tumourigenesis [72], and metastasis, thereby being recognized as a well-established CSC marker globally overexpressed in BCSCs [71]. In fact, several lines of evidence show that EpCAM might promote BCSCs survival through the activation of the Wnt signalling pathway [9,73,74]. Importantly enough, EpCAM has been widely recognised as being expressed in circulating tumour cells (CTCs) [75], which are highly suspected to be enriched in a CSC function and activity, allowing the survival of cancer cells in anoikis-independent conditions [76,77].

##### Cadherin 3

Cadherin 3, also called placental cadherin (P-cadherin), belongs to the classical cadherin family and plays a critical role in mediating cell–cell adhesion and structural integrity of epithelia, through the formation of cadherin–catenin complexes [78]. Its function as a mediator of stem cell properties during embryonic development is also recognised, as it is considered a biomarker for the isolation of stem cells. Due to its important biological role, it is expectable that alterations in P-cadherin are associated with disease, in particular in cancer [79]. Interestingly, P-cadherin expression correlates with poor patient’ prognosis both in primary breast tumours and in lymph node metastasis [80,81]. Moreover, studies from our group have shown that P-cadherin overexpression is associated with cell invasion, through the disruption of E-cadherin mediated cell–cell adhesion and through the modulation of *α*6*β*4 integrin-mediated cell-matrix adhesion [82,83,84]. Importantly, P-cadherin expression is also involved in stem-like properties and is associated with the expression of other CSC biomarkers, including CD44, CD49f, and ALDH1 [85]. In fact, our group has demonstrated that downregulation of P-cadherin was associated with reduced in vitro self-renewal ability and decreased in vivo tumourigenic potential [83]. Thus, P-cadherin is a valuable BCSC marker with relevant prognostic implications.

##### Erb-b2 Receptor Tyrosine Kinase 2

Erb-b2 receptor tyrosine kinase 2 (Erbb2/HER2) belongs to the human epidermal growth factor receptor family and is involved in the regulation of several pathways associated with cell division, proliferation, cell motility, invasion, differentiation, and apoptosis. In breast cancer, HER2 overexpression occurs in 25–30% of patients, composing a HER2-amplified breast cancer subtype, which is associated with a poor survival rate, as well a high risk of metastasis [86]. Emerging studies demonstrate that HER2 plays a role in the regulation of BCSC activity, in particular in self-renewal and radioresistance [87,88,89,90]. In fact, Duru et al. observed that HER2^+^ BCSCs isolated from HER2^-^ breast cancer cells showed to have enhanced ALDH1 activity, aggressiveness, and radioresistance [88].

##### Prominin 1

Prominin-1, also known as CD133, is a transmembrane glycoprotein commonly used as a stem cell marker due to its role in suppressing differentiation. In breast cancer, CD133 expression has been associated with poor overall survival probability, high tumour grade, lymph node metastasis, hormone receptor negativity, and HER2 positivity, as well as advanced tumours, nodes, and metastasis (TNM) stage [91]. Moreover, Joseph et al. demonstrated that the overexpression of CD133 was associated with a poor prognosis in invasive breast cancer [92]. The role of CD133 as a potential BCSCs’ biomarker was firstly described in a study by Wright et al., in which an increase in colony-forming efficiency, proliferative rate, and tumourigenic potential was observed in CD133^+^ cells derived from BRCA1 murine breast tumours [93]. Moreover, Croker et al. demonstrated that the ALDH^+^CD44^+^CD133^+^ population isolated from invasive breast cancer cell lines showed enhanced growth, colony formation, migration, and invasion capabilities, as well as greater in vitro and in vivo malignant and metastatic behaviour [94]. Additionally, Brugnoli et al. reported that CD133 silencing reduced the invasive capacities in triple-negative breast cancer cells [95].

##### C–X–C Motif Chemokine Receptor 1

C–X–C motif chemokine receptor 1 (CXCR1) is one of the receptors for CXC ligand 8 (CXCL8). In fact, the CXCL8–CXCR1 axis has been described as playing a crucial role in breast cancer stemness [96]. Interestingly, in a gene expression profile study, CXCR1 was found to be upregulated in ALDH^+^ BCSCs of various breast cancer cell lines [97]. In addition, treatment with recombinant IL-8 led to an enrichment in the BCSC population, since it resulted in an increased ALDH expression and mammosphere forming efficacy, as well as a greater invasive capacity [98]. Ginestier et al. reported that the inhibition of CXCR1 promoted the selective depletion of ALDH^+^ BCSCs through the FAK–AKT–FOXO3A pathway [97].

## 3. Clinical Relevance of BCSC Biomarkers

The high mortality associated with breast cancer is closely related to tumour recurrence and the development of metastatic disease. Importantly, BCSCs play crucial roles in these biological processes, since they are highly tumourigenic cells with self-renewal and differentiation capacities that are able to form or repopulate heterogeneous cancer cell populations. Moreover, BCSCs are highly resistant to conventional chemotherapies, and there may even be an enrichment in BCSCs population in post-treatment settings [6], thereby being mainly responsible for tumour recurrence [2]. In fact, in their clinical studies, Lee et al. demonstrated that there is an enrichment in CD44^+^CD24^−^ population in primary breast tumours following chemotherapy [99]. Accordingly, Creighton et al. reported an enrichment in the CD44^+^CD24^−^ population in residual tumour tissue after endocrine therapy, compared with pretreated samples, in a cohort of 52 breast cancer patients in phase II clinical trials [100]. Importantly, besides their role in chemoresistance and tumour recurrence, there are several clinical lines of evidence that suggest a relationship between BCSCs and metastasis. For instance, primary tumours from breast cancer patients with high CD44^+^CD24^−^ correlated with the presence of distant metastasis, including bone [101] and pleural metastasis [102], as well as show early bone marrow disseminated cancer cells [39]. In addition, ALDH1 expression, in particular the ALDH1A3 isoform, significantly correlated with aggressive clinical–pathological features [103] and distant metastases in inflammatory breast cancer [40], showing that CSC prevalence is directly associated with disease progression. More recently, we have studied the expression of five different BCSC markers (CD44, integrin subunit *α*6, cadherin-3, EpCAM, and ALDH1) and found significant enrichment for each biomarker in human breast cancer brain metastasis when compared with unmatched primary tumours. More importantly, our study revealed an enriched stem cell signature defined by the simultaneous expression of three to five of these BCSC markers. While the primary breast tumours only show this BCSC signature in 9.3% of the cases, this phenotype was clearly enriched in breast cancer brain metastasis, increasing the percentage to 55.6%. Even though a small proportion of the primary tumours display the BCSC signature, these were found to be significantly associated with poor clinical–pathological features and with decreased survival for breast cancer patients [42]. Thus, BCSCs have clinically relevant prognostic and therapeutic potential.

In this context, BCSC research in the metastatic setting is an urgent topic in oncology, which includes the identification of new BCSC biomarkers and models to study this population of clinical relevance. In fact, genomic approaches could be interesting to more accurately identify BCSC, as previously demonstrated [37]. Nonetheless, these methodologies are not as useful from a drug-targeting perspective, since they recognise a large number of genes that are not necessarily targetable as the cell surface biomarkers.

In this way, the development of BCSC-based tools that allow the prediction of prognosis and that can be translated into the clinic can constitute powerful approaches to ameliorate risk assessment and therapeutic selection of cancer patients. Notably, since there are difficulties in collecting serial biopsies from metastatic sites in patients, alternative biomarker-based methods should be considered. For instance, the search for BCSC biomarkers in circulating tumour cells could serve as surrogate evidence of anti-CSC activity, as already demonstrated by some authors [104,105].

In resume, the design of new therapeutic targeting-CSCs strategies could be a more effective anticancer therapy and might even allow therapy resistance to be overcome (Figure 2).

Importantly, the therapeutic targeting of BCSCs might be performed using different approaches, including cell surface, cytoplasmic and nuclear BCSC markers, stem cell-specific signalling pathways, or even BCSC microenvironmental factors. Immuno- and differentiation therapies might also be promising strategies to overcome BCSCs’ resistance to conventional therapies [32]. To date, there are already several BCSC-targeting therapies in clinical trials for different types of cancer, as previously reviewed by Saygin et al. [6], as well as by Yang et al. [5].

In the following sections, we will describe the BCSC-targeting therapies currently in clinical trials (Figure 3).

### 3.1. Targeting BCSC Signalling Pathways

#### 3.1.1. Notch Signalling

Notch signalling has been associated with BCSCs as playing a key role in promoting EMT, thereby contributing to CSC-mediated metastasis [17,106]. Evidence showed that inhibition of Notch signalling sensitises BCSCs to chemotherapy and radiation [107]. Importantly, Notch activation occurs after gamma-secretase cleavage of its intracellular domain. Thus, Notch-signalling-targeting studies have employed two different approaches: (1) gamma-secretase inhibition, in order to inhibit its role on Notch cleavage and consequent activation, and (2) the use of antibodies against Notch ligands or receptors. MK-0752 is a gamma-secretase inhibitor, already with a phase II clinical trial completed, that showed promising results targeting BCSCs, since a decrease in CD44^+^CD24^-^ population, as well as decreased mammosphere forming efficiency, was observed in samples from breast tumour serial biopsies [32,108]. Additionally, RO4929097 (RG-4733) and PF-03084014 (nirogacestat) are other gamma-secretase inhibitors already in clinical trials [31].

#### 3.1.2. Hh Signalling

Activation of Hh signalling has been associated with breast cancer since this pathway is crucial for tissue homeostasis and self-renewal. Importantly, the Hh pathway is abnormally upregulated in BCSCs, thereby promoting BCSCs proliferation [106]. Several Hh signalling pathway inhibitors have been developed for cancer treatment, targeting specifically the smoothened (SMO). Importantly, SMO is a transmembrane protein and a Hh ligand that mediates Hh downstream signalling pathway. Evidence showed that inhibition of Hh signalling, using cyclopamine, sensitizes BCSCs to paclitaxel [73]. In addition, sonidegib (LDE225) and vismodegib (GDC-0449), another SMO antagonist, are already in phase I clinical trials [31].

#### 3.1.3. Wnt Signalling

Increased activation of Wnt–*β*-catenin signalling significantly contributes to BCSCs’ self-renewal and therapy resistance [15]. Various studies showed that inhibition of the Wnt–*β*-catenin pathway suppressed BCSC proliferation and self-renewal [109]. Vantictumab is an antibody in phase I clinical trials, which inhibits Wnt–*β*-catenin signalling by blocking the Wnt receptor. Importantly, vantictumab has been showing interesting results, since it seems to reduce BCSC frequency and promote differentiation in patient-derived xenografts [110].

### 3.2. Targeting BCSCs’ Cell Surface Membrane Biomarkers

Membrane proteins are an area of special interest and account for more than 60% of current drug targets [55] and, therefore, are the major focus of this section. Importantly, most drugs achieve the therapeutic effect via interacting with membrane proteins, and thus, they represent major disease biomarker candidates and constitute promising therapeutic targets for monoclonal antibodies and other drugs, allowing the modulation of cell signalling through the inhibition of receptors or enzyme activity, for instance [55].

#### 3.2.1. CD44v6

CD44 exists in its standard form (CD44S), as well as in various isoforms (CD44v) due to post-translational modifications and alternative splicing of the primary transcript. CD44v6 is one of the CD44 isoforms and, in breast cancer, its overexpression in endocrine sensitive breast cancer cells has been connected to the activation of epidermal growth factor receptor (EGFR) signalling, to an increased invasive ability, as well as to a decrease in response to endocrine therapy [111,112,113]. Bivatuzumab mertansine is an immunoconjugate composed of a highly potent antimicrotubule agent and a monoclonal antibody against CD44v6. It has been tested in phase I clinical trials (NCT02254005, NCT02254031) in anthracycline and taxane-pretreated patients with metastatic breast cancer who were shown to have CD44v6 expression. In fact, bivatuzumab mertansine produced some interesting results, since it was able to target CD44v6 and to promote disease stabilisation in 50% of patients independently of dose level. However, due to a fatal toxic epidermal necrolysis that occurred in a parallel study, the clinical trials using this immunoconjugate have been discontinued [114].

#### 3.2.2. Epithelial Cell Adhesion Molecule (EpCAM)

Adecatumumab is a novel monoclonal antibody targeting EpCAM, which is already in phase II clinical trials. Unfortunately, adecatumumab treatment did not show an impact on objective tumour regression in EpCAM overexpressing metastatic breast cancer patients [115], highlighting the need for further investigations of this agent in patients with EpCAM-overexpressing tumours.

#### 3.2.3. Cadherin-3 (P-Cadherin)

P-cadherin constitutes an attractive target with therapeutic potential to treat tumours that highly express this protein, with several studies addressing its target using different approaches [116,117,118,119]. In fact, for breast cancer, PCA062, a P-cadherin-targeting antibody–drug conjugate may have clinical potential, since it has been shown to have potent antitumour activity with acceptable efficacy and tolerability in preclinical studies in primates [120]. However, PCA062 administration in phase I clinical trial (NCT02375958) has been discontinued, since the efficacy signals were limited during dose escalation [120], suggesting that further studies using this molecule need to be performed in order to successfully target P-cadherin. Moreover, further phase I clinical trials (NCT02454010) studies suggest the use of P-cadherin-targeted radioimmunotherapy with Y-FF-21101 monoclonal antibody in the treatment of solid tumours as showing promising results [119]; however, this treatment has not been tested in breast cancer patients, which might be an interesting approach for the future since P-cadherin overexpression is a common feature of breast cancer patients. PF-06671008 is a CD3-bispecific molecule-targeting P-cadherin that was able to promote T-cell-mediated regression of established tumours in mice models. Importantly, PF-06671008 is already in phase I clinical trials (NCT02659631) for triple-negative breast cancer patients [118,121].

#### 3.2.4. Erb-b2 Receptor Tyrosine Kinase 2 (Erbb2/HER2)

Increasing evidence demonstrates a connection between HER2 and BCSCs. Importantly, there are already well-established HER2-targeted therapies, such as trastuzumab, that clearly impact tumour recurrence and aggressiveness. Thus, it is plausible to assume that these HER2 inhibitors are also targeting HER2^+^ BCSCs, as demonstrated in recent and emerging studies. In particular, it has been demonstrated by Li et al. that neoadjuvant lapatinib treatment led to a decrease in the CD44^+^CD24^−^ population, as well as a reduction in mammosphere formation of samples from breast cancer patient biopsies [122]. Additionally, Korkaya et al. reported that treatment with trastuzumab reduced the BCSC population in HER2-overexpressing breast cancer cell lines [123].

#### 3.2.5. C–X–C Motif Chemokine Receptor 1 (CXCR1)

Repaxirin is an investigational inhibitor of CXCR1 that has been shown to reduce the CSC content of human breast cancer in mice xenografts [97], which is already in phase II clinical trials (NCT02370238, NCT01861054). The agent has shown promising results, promoting a decrease in CXCR1^+^ cells in serial biopsies following treatment. Additionally, a decrease in the BCSCs’ CD44^+^CD24^−^ and ALDH1^+^ populations was observed, with minimal adverse reactions registered in breast cancer patients [124].

## 4. Conclusions

There are several aspects of BCSC biology that need to be solved in order to successfully develop therapies that achieve BCSC elimination. Currently, a major limitation of the anti-BCSC therapies consists of their target, which is, mainly, stemness-associated transcriptional and/or intracellular signalling factors that are shared with normal stem cells and are also sometimes difficult to target. Moreover, the existing BCSC-signalling targeted therapies are not very specific, highlighting the need for the design of new inhibitors. Thus, it is of major importance to search for new potential biomarkers that could be easily translated to the clinical. The search for putative new CCSC biomarkers and the identification of novel and unique pathways that are active in this subset of cells is of clinical relevance. In particular, cell surface CSC markers seem to be among promising strategies since they constitute accessible targets for monoclonal antibodies or drugs, which allows easy modulation of CSC signalling.

## Figures and Tables

**Figure 1 cells-11-00934-f001:**
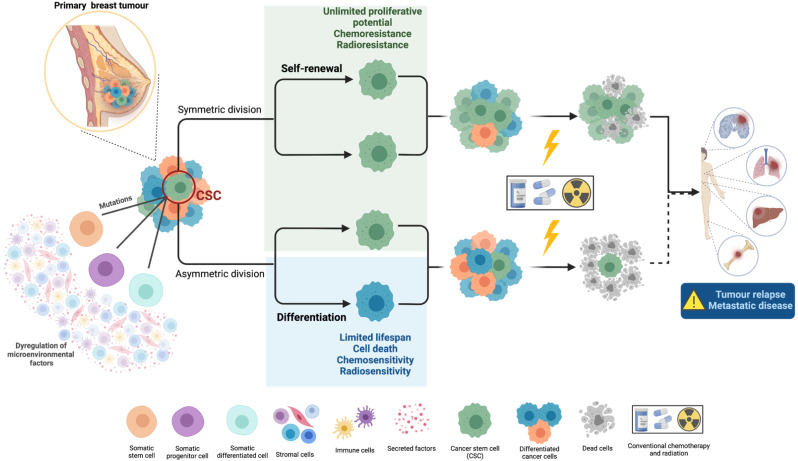
Main features of CSC biology: cell of origin, plasticity, and their role in metastasis. CSCs are characterized by their self-renewal and differentiation capacities, which confers them the ability to form/repopulate heterogeneous cancer cell populations. Moreover, due to their enhanced mechanisms of chemo- and radioresistance, CSCs are considered as the main responsible for tumour recurrence and metastases. Generally, CSCs divide asymmetrically, giving rise to a CSC and a differentiated cell that integrates with tumour bulk. However, under stress conditions, as cell loss during anticancer treatment, they can also divide symmetrically, originating two CSCs, which allows the excessive promotion of tumour growth and repopulation. Biorender: https://biorender.com (accessed on 12 January 2022).

**Figure 2 cells-11-00934-f002:**
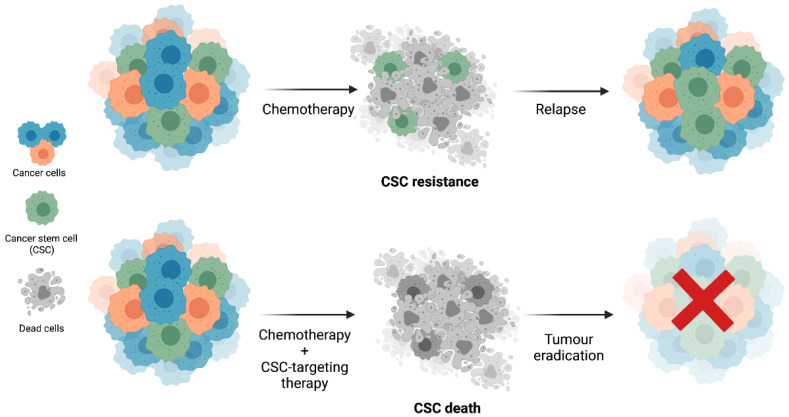
CSC resistance to conventional chemotherapy and the potential of CSC-targeting therapies for tumour eradication. The conventional chemotherapy that is currently in clinical practice targets highly proliferative cells and is ineffective against CSCs, which will ultimately be responsible for tumour relapses. The combination of standard chemotherapy with CSC-targeting therapies might be a promising strategy for tumour eradication. Biorender: https://biorender.com (accessed on 4 September 2021).

**Figure 3 cells-11-00934-f003:**
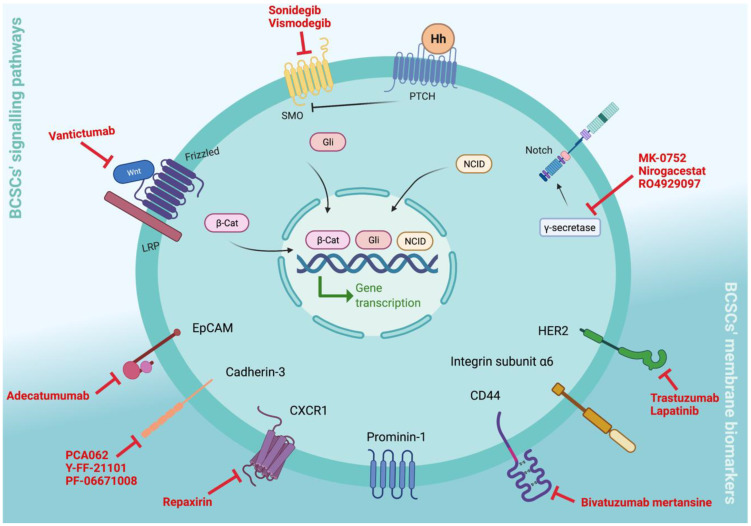
Breast cancer stem cell membrane biomarkers, signalling pathways, and targeting therapies in clinical trials. CD44, integrin alpha 6, EpCAM, Cadherin 3, HER2, prominin-1, and CXCR1 are commonly recognised as CSC membrane biomarkers, whose expression has been associated with breast cancer. Bivatuzumab mertansine is a CD44v6-targeted therapy that has been tested in phase I clinical trials in anthracycline and taxane pretreated patients with metastatic breast cancer who were shown to have CD44v6 expression. Adecatumumab is an EpCAM-targeting antibody already in phase II clinical trials; however, it does not have a significant impact on EpCAM overexpressing breast cancer patients. PCA062, Y-FF-21101, and PF-06671008 are molecules that specifically target cadherin-3 that are currently in clinical trials. Trastuzumab and lapatinib are drugs already in clinical practice as HER2-targeted therapies, which emerging studies describe as contributing to BCSC elimination. Repaxirin is in phase II clinical trials for CXCR1 inhibition, showing interesting results in the reduction in BCSC content. Since Wnt–*β*-catenin, Hh, and Notch signalling are associated with self-renewal, differentiation, apoptosis, and survival, they are usually upregulated in cancer and are, therefore, seen as promising for CSC-targeting therapies. In particular, for breast cancer, the CSC-targeting therapies in clinical trials have CSC signalling pathways as their main targets. Specifically, vantictumab is a Wnt inhibitor, sonidegib and vismodegib are SMO inhibitors, and MK-0752, nirogacestat, and RO4929097 are *γ*-secretase inhibitors. Biorender: https://biorender.com (accessed on 9 November 2021).

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
