# Peer review of "Breast Cancer Stem Cell Membrane Biomarkers: Therapy Targeting and Clinical Implications"

_cells, 2022, doi:10.3390/cells11060934_

Round 1
Reviewer 1 Report
Major comments:
The conceptual approach is superficial and does not provide sufficient insight. Overall, the information represented in this review manuscript is confusing and corresponds to over-lapping overviews on therapy-induced tumor dormancy and breast stem-like cancer cells. The objectivity is not clear. Confusing terminology is a major limitation, such as “breast cancer stem cell” or “breast stem-like cancer cell”.
In addition, several caveats of the presented review manuscript, as few described in the specific comments below, attenuate the significance of their concept and quality of the manuscript.
Other comments;
- The information in the manuscript is not cohesive and coherent.
- The title of the manuscript focused on targeting BCSC biomarkers. However, from the abstract and introduction section, I saw few words to describe BCSC or even BC. Even after reading half of the article, I started to see BCSC.
- What I learned from subtitle 2 is that role of CSC in metastasis should be predominantly discussed. However, it is simply mentioned in the last paragraph of this section.
- Line 21 “upon their first” what is their?? Nonspecific terminology and random statements are major drawbacks of this manuscript.
- Line 33 and 34: diverse differentiation direction instead of the proliferation of CSC is the direct reason for causing heterogeneity.
- Line 40 and 41: “constitutive NF-κB and EMT” do not meet the definition of a genetic or epigenetic alteration.
- Line 42 and 43: the authors should list the specific example of inflammation-induced CSC, instead of merely mentioning this issue.
- Line 44 and 45: the authors should explain what “distinct properties of CSC pools” are, and should give examples of such mentioned CSC pools, instead of simply mentioning them.
- Line 46 to 51: it is superficial to only mention CSC regulating factors of signalings without explaining how they regulate.
- Line 72 and 73: what is “significant prognostic and therapeutic implication”? It needs examples to further support.
- Line 76 and 77: what are specific “several markers”? Need examples.
- Line 81: “human metastatic lesion” and “metastatic cancer cells” need further explanation. The off-the-cuff introduction is unreasonable and superficial to claim “we have already described”.
- Line 96: epigenetic alteration can also result in CSC resistance to therapy.
- Line 89-98: extrinsic and intrinsic reasons of CSC resistance to therapy should be mentioned at first, then based on which to classify them.
- Line 118 and 199: Authors do not explain the relationship between “more efficient ROS-scavenging mechanisms and therapy resistance”. Simply mentioning ROS here makes readers confusing.
- Line 131: what kind of “abnormalities”? Need example.
- Line 135: what is an “important mechanism”? Need example.
- Line 136-138: Need specific example.
- Line 141 and 142: “Using human breast cancer samples” for what?
- Line 171 to 177: According to narrative logic, a variant of CD49f itself should be introduced first, then followed by interaction between CD49f with other proteins.
- Line 177 and 178: Which “signaling pathway” cooperates with CD49f dimer? be specific.
- Line 193 and 194: “role in mediating homophilic Ca2+ independent cell-cell adhesion” confusing!!!
- Subtitle 4 section: actually, widely accepted BCSC biomarkers are more than CD44, CD24, CD49f, ALDHA1, EpCAM, and p-cadherin. Why no introduction of other biomarkers??? What are the selection criteria?
- Line 218 to 220: “highly tumorigenic cells with self-renewal and differentiation capacities that are able to form or repopulate heterogeneous cancer cell populations” cannot lead to high mortality, but high risk of metastasis or relapse.
- Line 242: resistance to which therapy?
- Line 243 to 245: Since clinical trials, based approaches were reviewed by Saygin and Yang et al then what is the need for this review??
- Line 248 to 256: What is the relationship between gamma-secretase and Notch signaling? After the introduction of Notch signaling in BC, the author directly mentioned MK-0752, making readers confused about the connection.
- Line 257 to 262: No connection between Hh signaling and SMO.
- Line 307 and 308: since tumor microenvironment-based information is not discussed systematically then why sudden jump in concluding remarks? What is the basis?
- Nomenclature for the gene (mRNA and protein) should be represented as suggested by HUGO Gene Nomenclature Committee, or equivalent resources to ensure standardized nomenclature is used for species-specific gene and protein symbol/names.
- Viewpoint: Lacking perspectives and innovation.
- Here, I list some recently published review articles in the field of CSC (1-3), whose viewpoints are similar to this manuscript. However, knowledge of CSC in this article is not as abundant and systematic as those articles that I list. Herein, why would readers prefer this article since a more comprehensive CSC review article already existed?
- The English language of this article is not so appropriate or understandable. A few examples;
- Line 36: I am confused why using “if” here.
- Line 147 to 150: sentence doesn’t make sense.
- Line 151: should be “whose ligand”.
- Line 152: the position of the word “important” is strange in the sentence.
- Line 181: “belongs”.
- Line 181 to 184: a sentence is too complex to make readers understand.
- Line 204 and 205: objective changed from “them” to “it” make readers confused.
- Zeng X, et al. (2021) Breast cancer stem cells, heterogeneity, targeting therapies and therapeutic implications. Pharmacol Res 163:105320.
- Palomeras S, Ruiz-Martinez S, & Puig T (2018) Targeting Breast Cancer Stem Cells to Overcome Treatment Resistance. Molecules 23(9).
- Yang F, Xu J, Tang L, & Guan X (2017) Breast cancer stem cell: the roles and therapeutic implications. Cell Mol Life Sci 74(6):951-966.
Author Response
We thank the Reviewer 1 for the critical review and for the detailed comments. Below we provide a point-by-point response, including the clarification of the objective of this review, which focus on breast cancer stem cells membrane biomarkers. Importantly, we reviewed all the manuscript to standardize the terminology and to expand the revised information under this subject, in order to significantly improve its novelty and quality. All the changes made in the manuscript are highlighted in red.
Other comments:
The information in the manuscript is not cohesive and coherent.
We agree with Reviewer 1 that the manuscript was lacking cohesion. Thus, we have now profoundly revised the manuscript to make it clearer and more coherent. The reviewer can find all the modifications highlighted in red.
The title of the manuscript focused on targeting BCSC biomarkers. However, from the abstract and introduction section, I saw few words to describe BCSC or even BC. Even after reading half of the article, I started to see BCSC.
We thank Reviewer 1 for raising this point. We are aware that the title does not clearly demonstrate the main goal of this review. Thus, we have now clarified at the title, abstract and introduction section that our intent is to review the already described breast cancer stem cell membrane proteins as potential clinical and therapeutic biomarkers in breast cancer.
What I learned from subtitle 2 is that role of CSC in metastasis should be predominantly discussed. However, it is simply mentioned in the last paragraph of this section.
We thank Reviewer 1 for this suggestion. We have now expanded this topic in section 2, lines 157-167.
Line 21 “upon their first” what is their?? Nonspecific terminology and random statements are major drawbacks of this manuscript.
We understand the Reviewer’s concern and apologize for the nonspecific terminology. Therefore, we have revised all those random statements and deepen the current knowledge on those topics.
Line 33 and 34: diverse differentiation direction instead of the proliferation of CSC is the direct reason for causing heterogeneity.
We agree with Reviewer 1, and therefore, we clarified that point (see Line 44).
Line 40 and 41: “constitutive NF-κB and EMT” do not meet the definition of a genetic or epigenetic alteration.
We agree with Reviewer 1, and therefore, we removed that sentence for the manuscript (see Line 48-50).
Line 42 and 43: the authors should list the specific example of inflammation-induced CSC, instead of merely mentioning this issue.
We agree with Reviewer 1, and therefore, we clarified that point (see Line 50-55).
Line 44 and 45: the authors should explain what “distinct properties of CSC pools” are, and should give examples of such mentioned CSC pools, instead of simply mentioning them.
We agree with Reviewer 1, and therefore, we clarified that point (now described in section2, Lines 176-181).
Line 46 to 51: it is superficial to only mention CSC regulating factors of signaling without explaining how they regulate.
We thank the Reviewer for raising this point. We did not explore how these specific signaling pathways regulate the CSC function and activity, since it was not the focus of this specific review (and as already mentioned by the reviewers, this topic has been highly revised by other authors).
Line 72 and 73: what is “significant prognostic and therapeutic implication”? It needs examples to further support.
We agree with Reviewer 1, and therefore, we clarified that point (see Line 74-78).
Line 76 and 77: what are specific “several markers”? Need examples.
We agree with Reviewer 1 and, therefore, we clarified that point indicating that “CSCs can be also identified through the expression of several markers, mainly cell-surface proteins like CD44, CD133 and EpCAM” (see Line 81).
Line 81: “human metastatic lesion” and “metastatic cancer cells” need further explanation. The off-the-cuff introduction is unreasonable and superficial to claim “we have already described”.
We agree with Reviewer 1 and, therefore, we clarified that point. In detail, we changed this part of the introduction for section 2 about breast cancer stem cells. Also, we clarified the terms “human metastatic lesion” and “metastatic cancer cells” (see Line 206-210).
Line 96: epigenetic alteration can also result in CSC resistance to therapy.
We agree with Reviewer 1 and we have now clarified that point (see Line 97-105).
Line 89-98: extrinsic and intrinsic reasons of CSC resistance to therapy should be mentioned at first, then based on which to classify them.
We agree with Reviewer 1 and, therefore, we clarified that point (see Line 97-107).
Line 118 and 199: Authors do not explain the relationship between “more efficient ROS-scavenging mechanisms and therapy resistance”. Simply mentioning ROS here makes readers confusing.
We agree with Reviewer 1 and, therefore, we clarified that point (see Line 127-130).
Line 131: what kind of “abnormalities”? Need example.
As suggested by the Reviewer, we clarified this point (see Line 141-142).
Line 135: what is an “important mechanism”? Need example.
As suggested by the Reviewer, we gave examples on what we described as “important mechanism” (see Line 145-151).
Line 136-138: Need specific example.
As suggested by the Reviewer, we gave examples on what we described as “important mechanism” (see Line 151-154).
Line 141 and 142: “Using human breast cancer samples” for what?
As suggested by the Reviewer, we reformulated this part of the text (see Line 157-167).
Line 171 to 177: According to narrative logic, a variant of CD49f itself should be introduced first, then followed by interaction between CD49f with other proteins.
As suggested by the Reviewer, we have now introduced this information (see Line 285-294).
Line 177 and 178: Which “signaling pathway” cooperates with CD49f dimer? be specific.
As suggested by the Reviewer, we have now clarified this point (see Line 285-294).
Line 193 and 194: “role in mediating homophilic Ca2+ independent cell-cell adhesion” confusing!!!
We agree with the Reviewer that this statement was confusing and, therefore, we clarified it in the new version of the manuscript (see Line 298-299).
Subtitle 4 section: actually, widely accepted BCSC biomarkers are more than CD44, CD24, CD49f, ALDHA1, EpCAM, and p-cadherin. Why no introduction of other biomarkers??? What are the selection criteria?
We thank the Reviewer for raising this point and have now reformulated this section (currently is section 2.1.) and clarified the selection criteria for the abovementioned markers. Importantly, we excluded ALDH1 and added new information on other BCSC membrane proteins that are clear therapeutic targets in breast cancer. This new information can be found in section 2.1 Line 218-365).
Line 218 to 220: “highly tumorigenic cells with self-renewal and differentiation capacities that are able to form or repopulate heterogeneous cancer cell populations” cannot lead to high mortality, but high risk of metastasis or relapse.
As suggested by the Reviewer, we have now clarified this point (see 370-374).
Line 242: resistance to which therapy?
We thank the Reviewer for the comment and we have now included that information (see Line 425).
Line 243 to 245: Since clinical trials, based approaches were reviewed by Saygin and Yang et al then what is the need for this review??
We agree that Saygin and Yang did a profound revision clinical trial-based approach, mainly focusing on the clinical significance of CSCs and on CSC-targeting therapies in several cancers. In this revised version, we have clearly reinforced our focus on BCSC membrane molecules as potential targets for therapy, specifically in breast cancer.
Line 248 to 256: What is the relationship between gamma-secretase and Notch signaling? After the introduction of Notch signaling in BC, the author directly mentioned MK-0752, making readers confused about the connection.
As suggested by the Reviewer, we have now clarified the relationship between gamma-secretase and Notch signaling (see Line 435-439).
Line 257 to 262: No connection between Hh signaling and SMO.
As suggested by the Reviewer, we have now clarified this point (see Line 447-456).
Line 307 and 308: since tumor microenvironment-based information is not discussed systematically then why sudden jump in concluding remarks? What is the basis?
We agree with Reviewer 1 that the mention on tumor microenvironment in the concluding remarks is unexpected. We have now reformulated this conclusion session (Line 559).
Nomenclature for the gene (mRNA and protein) should be represented as suggested by HUGO Gene Nomenclature Committee, or equivalent resources to ensure standardized nomenclature is used for species-specific gene and protein symbol/names.
We agree that all the document should be standardized for gene and protein nomenclature. Thus, we have revised the complete manuscript.
Viewpoint: Lacking perspectives and innovation.
Here, I list some recently published review articles in the field of CSC (1-3), whose viewpoints are similar to this manuscript. However, knowledge of CSC in this article is not as abundant and systematic as those articles that I list. Herein, why would readers prefer this article since a more comprehensive CSC review article already existed?
We thank Reviewer 1 for raising this point. We are aware that the initial manuscript did not reflect the main goal of this review. Thus, we have now clarified it and revised all the document, focusing on breast cancer stem cell membrane proteins, as potential clinical and therapeutic biomarkers in breast cancer. As also mentioned, this review is not as systematic as those review articles; nevertheless, we are reviewing a new topic that has never been described in these other manuscripts. We are conscious that our first version was lacking that novelty, but this new version is far more complete and focused on our topic of interest.
Considering the reviews mentioned by Reviewer 1:
- Zeng X et al. (2021) explores the characteristics of BCSCs and the roles of BCSCs in the formation, maintenance and recurrence of breast cancer, the self-renewal signaling pathways in BCSCs, the BCSC microenvironment, potential therapeutic targets related to BCSCs and current therapies and clinical trials targeting BCSCs. Importantly, in our revised version there is current information on several other BSCS membrane biomarkers that are not described in this manuscript, such as CD44v6, EpCAM and Cadherin-3.
- Palomeras S et al. (2018) reviews BCSCs with future directions in the establishment of a therapy targeting this population. They also summarize drugs targeting the main BCSCs’ signaling pathways undergoing clinical trials, lacking once again several other important BCSC membrane biomarkers.
- Yang et al. (2017) summarize extensively the current literature about the diversity of BCSC markers, the roles of BCSCs in tumor development and the regulatory mechanisms of BCSCs. Nevertheless, the authors do not explore the impact and current targeting therapies in BSCS membrane biomarkers.
The English language of this article is not so appropriate or understandable. A few examples;
- Line 36: I am confused why using “if” here.
- Line 147 to 150: sentence doesn’t make sense.
- Line 151: should be “whose ligand”.
- Line 152: the position of the word “important” is strange in the sentence.
- Line 181: “belongs”.
- Line 181 to 184: a sentence is too complex to make readers understand.
- Line 204 and 205: objective changed from “them” to “it” make readers confused.
We followed the Reviewer’ suggestion and have revised the entire manuscript for English language.

Reviewer 2 Report
The review manuscript by Conde and colleagues depicts a fair portrait of the current situation with CSC biomarkers and CSC therapeutics in breast cancer.
There is no evidence that any of the interventions tested so far in early phase clinical trials up to randomized phase 2 studies provide clinical benefit, thus casting doubts on the validity of CSC as a therapeutic target.
The manuscript is well organized, however, in this reviewer’s opinion, it lacks originality and it does not address one key question which is inherent in the article title: are breast cancer CSC markers one of the factors why CSC targeted therapies have not delivered thus far?
The section “Breast cancer stem cells” should highlight if, and to what extent, any given marker is expressed by any normal cell, in particular normal breast cells (reference 46 of the article for ALDH) . This is stated in the conclusions only, but it could be explained in more details in the abovementioned section, including the possibility that stemness is a transient state of some cancer cells which may transition to a differentiated cancer cell and then back to a CSC state (please refer to Garber K Nature Review Drug Discovery volume 17 November 2018).
The section “Clinical relevance of breast cancer stem cells biomarkers” should discuss in more details what the relationship is between a given CSC survival pathway, a corresponding targeted therapy and any CSC biomarker. For example, is there robust evidence of CSC (identified by any biomarker) reduction in paired clinical samples before and after treatment in any breast cancer type? Any suggestions for future clinical investigations? For example:
- Could gene signature be used to more precisely identify breast cancer CSC than the biomarkers described in the article?
- How many primary breast cancer samples stain positive for any CSC biomarker?
- How many metastatic samples stain positive?
- Why do some samples even in primary breast cancer stain negative for any CSC biomarker? is it sampling bias?
- Considering the difficulties in collecting serial biopsies from metastatic sites in patients, it should be discussed if alternative biomarker-based methods (e.g., circulating CSC?) can serve as a surrogate evidence of anti-CSC activity
Since the review is focused on breast cancer, there are CSC-targeted approaches not mentioned in this article that completed randomized clinical trials, such as targeting of CXCR1, and publications about targeting of HER-2+ breast cancer CSC by trastuzumab and lapatinib. Also, work from Max Wicha’s group highlighted some differences between CD24-/CD44+ and ALDH-1+ breast cancer CSC that are not mentioned in the manuscript
Author Response
We thank Reviewer 2 for raising this point. As mentioned to Reviewer 1, we are aware that the initial manuscript did not reflect the main goal of this review. Thus, we have now clarified it and revised all the document, focusing on breast cancer stem cell membrane proteins as potential clinical and therapeutic biomarkers in breast cancer. We are conscious that our first version was lacking novelty, but this new version is more focused on our topic of interest. Importantly, it is our conviction that one of the main reasons for the lack of efficiency of CSC targeted therapies is related with the difficult access for identification and targeting of CSC biomarkers. Therefore, the commonly used therapies to transcription factors and signaling pathways may be one of the reasons for the lack of success. We do think that focusing on cell membrane proteins that are enriched in cancer stem cell populations should be a better strategy to overcome these disappointing clinical results. Nonetheless, for these biomarkers, there are less available information.
The section “Breast cancer stem cells” should highlight if, and to what extent, any given marker is expressed by any normal cell, in particular normal breast cells (reference 46 of the article for ALDH1). This is stated in the conclusions only, but it could be explained in more detail in the abovementioned section, including the possibility that stemness is a transient state of some cancer cells which may transition to a differentiated cancer cell and then back to a CSC state (please refer to Garber K Nature Review Drug Discovery volume 17 November 2018).
We thank the Reviewer for the comment and suggestion. We have now clarified that point and added this information to the revised version (please see Line 176-184).
The section “Clinical relevance of breast cancer stem cells biomarkers” should discuss in more details what the relationship is between a given CSC survival pathway, a corresponding targeted therapy and any CSC biomarker. For example, is there robust evidence of CSC (identified by any biomarker) reduction in paired clinical samples before and after treatment in any breast cancer type? Any suggestions for future clinical investigations? For example:
Could gene signature be used to more precisely identify breast cancer CSC than the biomarkers described in the article?
How many primary breast cancer samples stain positive for any CSC biomarker?
How many metastatic samples stain positive?
Why do some samples even in primary breast cancer stain negative for any CSC biomarker? Is it sampling bias?
Considering the difficulties in collecting serial biopsies from metastatic sites in patients, it should be discussed if alternative biomarker-based methods (e.g., circulating CSC?) can serve as a surrogate evidence of anti-CSC activity
We thank the Reviewer for the comment and for the suggestion. We agree that these are important points that should be mentioned. Thus, we have now introduced and discussed these topics in the reformulated version of the manuscript (Section 2, Line 366-397).
Since the review is focused on breast cancer, there are CSC-targeted approaches not mentioned in this article that completed randomized clinical trials, such as targeting of CXCR1, and publications about targeting of HER-2+ breast cancer CSC by trastuzumab and lapatinib. Also, work from Max Wicha’s group highlighted some differences between CD24-/CD44+ and ALDH-1+ breast cancer CSC that are not mentioned in the manuscript
We thank the Reviewer for the comment and suggestion. In fact, that important information was lacking from the initial version of this manuscript. Therefore, we have now included information on these BSCS membrane proteins as biomarkers for clinical application (section 2.1.2, line 327-365; section 3.2, lines 515-532).

Round 2
Reviewer 1 Report
The authors precisely addressed all comments raised by the reviewer. I suggest proofreading the manuscript for any potential plagiarism.
Reviewer 2 Report
The revised article is a comprehensive review of cell surface markers for cancer stem cells. As such, it is more focused than the original submission and, in this reviewer's opinion, has gained originality and usefulness to the journal's readership.
I have no further suggestions or comments